# Novel Amperometric Mercury-Selective Sensor Based on Organic Chelator Ionophore

**DOI:** 10.3390/molecules28062809

**Published:** 2023-03-20

**Authors:** Basant Elsebai, Mariana Emilia Ghica, Mohammed Nooredeen Abbas, Christopher M. A. Brett

**Affiliations:** 1Water Pollution Research Department, Environmental and Climate Changes Research Institute, National Research Centre, El-Buhouth St., Dokki, Giza 12622, Egypt; 2Department of Chemistry, CEMMPRE, ARISE, University of Coimbra, 3004-535 Coimbra, Portugal; 3Department of Chemical Engineering, CIEPQPF, University of Coimbra, 3030-790 Coimbra, Portugal; 4Applied Organic Chemistry Department, Chemical Industries Research Institute, National Research Centre, El-Buhouth St., Dokki, Giza 12622, Egypt

**Keywords:** amperometric sensor, ionophore, N, N di (2-hydroxy-5-[(4-nitrophenyl)diazenyl]benzaldehyde) benzene-1,2-diamine, mercury, carbon nanotubes

## Abstract

A novel amperometric sensor for the direct determination of toxic mercury ions, Hg^2+^, based on the organic chelator ionophore N, N di (2-hydroxy-5-[(4-nitrophenyl)diazenyl]benzaldehyde) benzene-1,2-diamine (NDBD), and multiwalled carbon nanotubes (MWCNT) immobilized on a glassy carbon electrode surface was developed. The parameters influencing sensor performance including the ionophore concentration, the applied potential, and electrolyte pH were optimized. The sensor response to Hg^2+^ was linear between 1–25 µM with a limit of detection of 60 nM. Interferences from other heavy metal ions were evaluated and the sensor showed excellent selectivity towards Hg^2+^. The method was successfully applied to the determination of mercury ions in milk and water samples.

## 1. Introduction

Mercury is one of the widely found toxic elements in the environment, because it is highly reactive, extremely volatile, and relatively soluble in water and living tissues [1,2]. The threshold limits of mercury as a poisonous heavy metal in drinking water as defined by the United States Environmental Protection Agency (EPA) and the World Health Organization (WHO) are 10 and 5 nM, respectively [3]. High exposure to mercury can cause serious health problems such as kidney and respiratory failure, damage to the gastrointestinal tract and nervous system, failure of speech and hearing, reproductive toxicity, and even death [2,4]. Therefore, determination of this ion is very important.

Analytical methods reported for mercury ion determination in environmental and biological samples include [5,6,7]: cold vapour (CV) atomic fluorescence, CV coupled with atomic absorption or with an integrated quartz crystal microbalance, gas chromatography (GC) coupled with mass spectrometry, neutron activation analysis, inductively coupled plasma mass spectrometry, and colorimetry. These techniques require sample pre-treatment, expensive instrumentation, complicated devices, and skilled operators; thence, they are not suitable for on-site testing and monitoring. A simple new technology is needed with appropriate high capability to monitor mercury with a fast response, while being inexpensive and making on-site monitoring possible.

Carbon nanotubes (CNT) have become attractive for use in sensors since their first fabrication in 1991 [8]; they possess useful characteristic properties such as good electrical conductivity, high electrocatalytic effect, strong adsorptive ability, and excellent biocompatibility [9,10]. They are able to promote fast electron transfer, resulting in wide application in electrochemistry [11,12], among them sensors based on CNT-modified electrodes for mercury (II) determination [13,14]. Beside CNT, various modified electrodes have received increasing attention for enhancing the selectivity and sensitivity of mercury electrochemical measurements; a summary of these studies is given in [3,15]. The most used modifiers include metal nanoparticles and films [16,17], polymers [18,19], ion-imprinted polymers [20], and organic chelators/ionophores [21,22,23], such as what is proposed here. Several studies included a combination of these modifiers with carbon nanotubes [9,24,25,26]; however, in the case of chelators/ionophores, these nanocomposites were mostly used in potentiometric and ion-selective electrodes (ISE) [13,27,28]. As far as the literature survey could reveal, only one sensor with this combination was used in voltammetric detection of Hg [21], and there is no study using electrodes modified with CNT and organic chelator/ionophore as amperometric sensors.

The aim of the present work was to investigate the amperometric determination of Hg (II) by using glassy carbon electrodes modified with the novel ionophore [N, N di (2-hydroxy-5-[(4-nitrophenyl)diazenyl]benzaldehyde) benzene-1,2-diamine)], NDBD. The study focuses on the development of a relatively inexpensive and simple procedure to selectively determine mercury ions with a low detection limit, using this organic chelator, which has not been previously employed in electrochemical detection. The possibility of including multiwalled carbon nanotubes (MWCNT) together with NDBD in the sensor platform was also evaluated and the performance compared with NDBD modifier by itself. Optimization of different experimental conditions, determination of figures of merit, and comparison with the literature is carried out and discussed. Practical application to milk and water samples is described.

## 2. Results and Discussion

### 2.1. Structural, Morphological, and Chemical Characterization of Nanostructures

The morphological structure of the materials was investigated by transmission electron microscopy (TEM). The nature of the chemical bonds in the matrix was evaluated by Fourier-transform infrared (FTIR) spectroscopy and the crystallinity was studied by X-ray diffraction (XRD).

TEM images of NDBD/MWCNT were recorded and compared with pure MWCNT, Figure 1. In Figure 1A, it is possible to observe MWCNT bundles with hollow internal channels (Figure 1B) as previously reported [29], having different lengths and with diameters around 30 nm, as specified by the supplier. After modification by NDBD (Figure 1C), the morphology was maintained; however, it is clearly observed that the tubes are thicker, and several structures are visible around the tubes (indicated by the red arrows), which may possibly be attributed to the presence of ionophore moieties, and in agreement with other studies, which attributed similar observations to the presence of an organic compound, such as a polymer [30].

Figure 2A depicts the XRD patterns of the MWCNT, ionophore, and MWCNT modified with the ionophore. Although MWCNT is considered a non-crystalline material, XRD has been successfully used to obtain information about its structural features [29,31]. In our study, it was possible to observe two characteristic diffraction peaks of carbon nanotubes, attributed to the plane reflection (002) and (100) of carbon. They appear at 2θ = ~25.9°, and at 2θ = ~42.7°, as reported also in [29,31], and correspond to the periodicity between the graphene layers and within the graphene layers. Regarding the ionophore, the XRD pattern also exhibits a wide peak around 2θ = ~26°, corresponding to plane reflection (002) of carbon, as well as two other peaks at ~15° and 17°, which may be attributed to the nitroaniline and the phenylenediamine moieties forming the ionophore’s structure [32,33]. These last two peaks are smaller in intensity in the case of NDBD/MWCNT, probably because the nanotubes partially cover the NDBD. No shift in the peak positions was observed, indicating a physical, rather than chemical, interaction.

FTIR spectra of the MWCNT, ionophore, and MWCNT modified with ionophore are shown in Figure 2B. Regarding the MWCNT spectra, the main bands observed may be attributed as follows: at 3307 cm^−1^ is the O-H stretch from carboxyl groups; at 2925 and 2855 cm^−1^ is the asymmetric/symmetric stretching vibration of H-H bonds from long alkyl chains; at 1589 cm^−1^ the carboxylate anion stretch; and at 1655 cm^−1^, the C=C stretch [34]. The spectrum of the ionophore shows the following bands characteristic of the main groups present on the organic backbone: between 3543–3211 cm^−1^, the typical broad stretching vibration band of O-H from the phenol; at 1620 cm^−1^, the stretching of the imine group (C=N); at 1580 cm^−1^, the stretching vibration of the azo group (N=N); at 1508 and 1340 cm^−1^, the asymmetric and symmetric stretching of (N-O) in nitro (NO_2_) group, attached to the aromatic ring. Between 1500 and 400 cm^−1^, a complex set of overlapping vibration bands is observed, considered the fingerprint region of phenol, in particular the characteristic stretching vibration of C-O in phenol at 1289 cm^−1^. These results are in agreement with those observed in [35,36,37] for other azo–azomethine compounds. Most of these bands disappear for NDBD/MWCNT, as well as a slight shift by ~15 cm^−1^ in the C=C band. This might be attributed to a possible π–π interaction between ionophore and MWCNT, in addition to physical linking. An increase in the -OH stretching band is also observed, due to the presence of carboxylic groups in oxidized nanotubes.

### 2.2. Electrochemical Characterisation of the Modified Electrodes in the Presence of Hg^2+^

The cyclic voltammetry of different electrodes (GCE, NDBD/GCE, MWCNT/GCE, NDBD/MWCNT/GCE) in the presence of 2.4 mM Hg^2+^ in 0.1 M acetate buffer, pH 4.0 is illustrated in Figure 3A. As can be observed, in all cases, a redox couple appears around +0.33 V for oxidation and +0.13 V for reduction, ascribed to the quasi-reversible Hg electrochemical process with the transfer of two electrons [37,38], corresponding to ${Hg}^{0}-2e^{-}↔{Hg}^{2+}$. A large increase in current peak with each modification step is visible, NDBD/MWCNT/GCE exhibiting the highest current. In all cases, an increase in peak potential separation is verified on increasing the scan rate, as shown in Figure 3B for NDBD/MWCNT/GCE.

### 2.3. Optimization of Fixed-Potential Amperometry Experimental Conditions

Amperometric measurements at the modified electrode were performed by successive injections of mercury (II) cations into stirred acetate buffer solution. Experimental parameters that can influence the performance, namely, the applied potential, the pH of the supporting electrolyte, and the ionophore concentration, were studied in order to optimize the response to mercury ions.

#### 2.3.1. Influence of the Applied Potential

The influence of the applied potential in the range of −0.3 V to +0.4 V vs. Ag/AgCl was investigated by comparing the sensor response to the same Hg^2+^ concentration, and the results are illustrated in Figure 4. The current response to Hg^2+^ slightly decreases from −0.3 to 0.0 V, then begins to increase at positive values, up to +0.2 V, where the maximum is obtained, and finally a large decrease in the response occurs at +0.3 and +0.4 V. This behavior can be explained considering the electrochemical processes occurring at the modified electrode in the presence of mercury ions (see Figure 3) as discussed above. The decrease in current response by amperometry occurs at a slightly lower potential than the peak observed by cyclic voltammetry. This is not unexpected, since CV is dynamic so that the fixed potential amperometry peak will correspond to a CV obtained at low scan rate. As observed in Figure 3B, when the scan rate is increased, the CV peak potentials shift more apart. The response is the highest at +0.2 V, an increase in current of 30% occurring from 0.0 V to +0.1 V; from +0.1 V to +0.2 V, the current increases by a further 10%, which can be correlated with the ionophore attracting Hg^2+^ and making a stable complex. A possible explanation of the decrease at higher potentials from +0.3 V onwards may be that the ionophore complex is less stable. It was decided to choose +0.1 V as the applied potential in fixed potential amperometry as being sufficient to obtain a well-defined response. The reason for choosing this potential that is closer to 0 V is to reduce the effect of possible interferences from other ions, which undergo reaction around +0.2–0.3 V, e.g., Cu(II) [39].

#### 2.3.2. Influence of pH

The sensitivity of the modified electrode can depend significantly on the pH of the medium. The influence of the pH of the supporting electrolyte on the response at NDBD/GCE at pH values between 3.0 and 5.0 was investigated, as shown in Figure 5. An increase in the response to mercury ions from pH 3.0 to pH 4.0 is observed, then, at pH 5.0, the response decreases. The decrease above pH 4.0 may be attributed to the formation of hydroxy complexes of Hg (II), as also observed in other studies, although the pH at which this occurs varies according to the modifier used. For example, in [1] this decrease was observed at pH higher than 4.0, in [25] it happened above pH 3.0, while in [18] it was only seen at pH values higher than 7.0. On the other hand, lower pH values lead to competition between mercury ions and protons [1]. Since in this work the highest response was achieved at pH 4.0, this was selected for further mercury ion amperometric measurements.

#### 2.3.3. Influence of the Ionophore Concentration

The amount of immobilised ionophore was varied in order to evaluate how much led to the best response towards mercury cations. Different concentrations, namely, 3, 5, 7, and 9 mg mL^−1^ of ionophore solution (0.3, 0.5, 0.7, and 0.9%) were used to modify the electrode surface and the calibration curves for Hg^2+^, see Figure 6, were all constructed under the same conditions. The response towards mercury cations increases when the ionophore loading is increased from 3 to 5 mg mL^−1^, then is lower for 7 and 9 mg mL^−1^; see Table 1. This can be explained by higher loadings of ionophore leading to thicker covering membranes, which limits the diffusion of Hg^2+^ through the modifier layer, hence, decreasing the sensitivity. At 5 mg mL^−1^ NDBD, the highest sensitivity and the lowest detection limit are achieved. Thus, a concentration of 0.5% ionophore was selected.

As seen from Table 1, in 0.1 M acetate buffer, pH 4.0, applied potential +0.1 V vs. Ag/AgCl and the best ionophore concentration of 0.5%, mercury can be determined at NDBD/GCE with a linear range from 150 to 1250 µM, a sensitivity of 55.0 µA cm^−2^ µM^−1^, and a detection limit (3X standard deviation of the regression line/slope) [40] of 11.6 µM.

### 2.4. Determination of Mercury Cations at Ionophore Modified Electrodes—Influence of MWCNT

Nanomaterials can increase the active surface area of modified electrodes and may exhibit electrocatalytic effects in the determination of various analytes. Carbon nanomaterials are good alternatives to conventional adsorbents for heavy metals owing to their high surface area [41]. However, carbon nanomaterials do not show a high affinity for mercury [42], so their combination with other materials with complexing capabilities is needed. Thus, the effect of modification of GCE with MWCNT before covering with ionophore was evaluated, using the better concentration of NDBD tested, i.e., 5 mg mL^−1^. Two different concentrations of MWCNT were tested: 0.2 and 0.5% *w*/*v*, Figure 7.

With 0.2% MWCNT, no increase in sensitivity for Hg^2+^ is obtained, but there is a decrease in the detection limit by a factor of 7 to 1.7 µM. For 0.5% MWCNT, a large increase in sensitivity, by almost 10 times, is achieved and the detection limit is sub micromolar, 0.06 µM. For both MWCNT loadings tested, a decrease in the upper limit of the linear range for the detection of mercury ions at NDBD/MWCNT/GCE (Table 2) is observed compared with that at NDBD/GCE (Table 1); however, this does not represent a problem for real sample measurements.

In Table 3, a comparison of the analytical performance of mercury detection of the sensor developed here with those in the literature is presented. The purpose was to compare with similar architectures (carbon nanotubes + other modifier, preferentially ionophore), independently of the electrochemical detection method. However, it is difficult to make a fair comparison with the literature, since the sensors containing both MWCNT and ionophore were potentiometric. Most of the voltammetric sensors use anodic stripping voltammetry, which allows detection at lower concentrations owing to the pre-concentration; thus, the detection limit obtained in this work is not as low. Nevertheless, the detection limit achieved here was much lower than that in [43,44], which also used preconcentration. Furthermore, the sensor developed by us has the advantage of a simpler platform than in [9,27,44]. On the other hand, only two sensors used fixed potential techniques for mercury measurement and their architecture does not include carbon nanotubes or ionophores. In [17], where chronoamperometry was used for detection, a much higher fixed potential than here was applied, +0.65 V vs. SCE, and is based on the formation of a redox inactive complex with thiols, L-cysteine and 1,4-butanedithiol, and is, thus, an indirect method for mercury detection; the detection limit is higher than here. In [23], a very high potential was applied, +0.9 V vs. SCE; the detection limit was not mentioned, but mercury was measured in a similar linear range as here.

### 2.5. Repeatability, Reproducibility, and Stability

The repeatability of the NDBD/MWCNT/GCE was investigated by recording four consecutive calibration curves for Hg^2+^ and comparing the sensitivities; the relative standard deviation, RSD, was 1.6%. The reproducibility with three different electrodes was 3.0%. The storage stability was assessed by keeping the modified electrode dry at room temperature and performing a calibration curve once a week; after 4 weeks, a decrease of 10% was observed. This is better than that of MB/AuNP/MWCNT/PANI/ITO in [44], which lost half of the activity after 3 weeks. Other sensors lost 5.5% of the response after one week with daily use [26], and “no major change” after one month storage was observed for MWCNT-CQD/GCE [45].

### 2.6. Selectivity

Selectivity with respect to different interferents, including the cations Pb^2+^, Cd^2+^, Cu^2+^, Co^2+^, Ni^2+^, Mn^2+^, and Cr^3+^ and the anions PO_4_^3−^ and Cl^−^ was studied (Figure 8), in a ratio of 1:2 of mercury to interferent (3 µM Hg^2+^ and 6 µM interferent ions). Most of these species did not lead to any change in the response of the electrode. Only copper and manganese ions exhibit a small interference, 3.7% Cu^2+^ and 6.3% Mn^2+^ (Figure 8B), thus, demonstrating excellent selectivity of the novel developed electrode for the determination of mercury ions in natural samples. Interference by Cu^2+^ was also observed in [17]. The selectivity depends on the strength of the ion–ionophore interaction, which occurs through binding with nitrogen and oxygen [28]. Excellent selectivity for mercury was also observed in other studies based on ionophores [27,28] and may be explained considering the extraordinary stability of N–Hg^II^–N bonding [47].

### 2.7. Application

To demonstrate the feasibility of the modified electrode for food and environmental use, application to the determination of Hg^2+^ in spiked milk and spiked tap water by the standard addition method was carried out. Each measurement was performed in triplicate and consisted of four additions: the first was the spiked water/milk sample and this was followed by three additions of 5 µM standard mercury ion solution. Taking into consideration the linear range of the sensor, the spiking of the samples was performed with three different concentrations of mercury (2, 5, and 10 µM), each used in separate measurement sequences. The recoveries were in the range of 93.5 to 98.1% (Table 4), indicating high efficacy of the electrode for practical analysis.

## 3. Materials and Methods

### 3.1. Reagents

All reagents were of analytical grade and were used without further purification. Multiwalled carbon nanotubes (MWCNT) with ~95% purity, 30 ± 10 nm diameter, and 1–5 µm length were from NanoLab, Waltham, MA, USA. 4-nitroaniline, salicylaldehyde, o-phenylenediamine, chitosan (Chit) of low molecular weight with a degree of deacetylation of 80%, ethanol, toluene, diethyl ether, and acetic acid were acquired from Sigma-Aldrich. Sodium nitrite, sodium acetate, and trisodium citrate were obtained from Merck. Acetate buffer solutions of various pH values were prepared by mixing standard stock solutions of 0.1 M sodium acetate and 0.1 M acetic acid and adjusting the pH with HCl or NaOH, both from Riedel De Haën, Germany. For the measurement of Hg (II), the appropriate amount of Hg(NO_3_)_2_ (Fisher Scientific, Waltham, MA, USA) was dissolved in water.

All solutions were prepared with Millipore MilliQ ultrapure water (resistivity > 18 MΩ cm) and experiments were performed at room temperature (25 ± 1 °C).

### 3.2. Instrumentation

Amperometric and voltammetric experiments were performed with an Ivium CompactStat potentiostat (Ivium Technologies B.V., Eindhoven, Netherlands). All electrochemical measurements were carried out at room temperature in a conventional three-electrode cell containing a bare or modified glassy carbon electrode (GCE) (EDAQ, ET074-3 Glassy Carbon Disk Electrode), with a diameter of 1 mm, as working electrode, a platinum wire as auxiliary electrode, and a Ag/AgCl (3 M KCl) electrode (Metrohm-Autolab) as reference.

The functional groups present in the ionophore were identified by Fourier-transform infrared spectroscopy using a Jasco (FT/IR 6100), USA.

The morphology of the modified electrode surface was examined using a JEOL JEM-1230 transmission electron microscope (TEM) with an acceleration voltage of 10 kV.

The crystalline structure of the NDBD ionophore, MWCNT, and NDBD/MWCNT was analyzed with powder X-ray diffraction (XRD), using a Bruker diffractometer, Bruker D 8 advance target. The radiation source of the diffractometer was CuKα with a second monochromator of a wavelength equal to 1.5405 Ǻ.

### 3.3. Preparation of [N, N Di (2-hydroxy-5-[(4-nitrophenyl)diazenyl]benzaldehyde) benzene-1,2-diamine) (NDBD)

First, 4-nitroaniline (30 mmol L^−1^) was dissolved in 18% hydrochloride solution (15 mL) at 0–5 °C. The reaction flask was immersed in an ice-bath for temperature control. Sodium nitrite (31 mmol L^−1^) was dissolved in 15 mL cold water and added dropwise to the reaction mixture during 30 min of stirring. Diazonium salt was obtained and used for coupling to salicylaldehyde, as reported in [35].

Salicylaldehyde (30 mmol L^−1^) was added to the cold 10% NaOH solution (15 mL) in a three-necked flask immersed in an ice-bath. Freshly prepared diazonium salt (0–5 °C) was added dropwise for 1 h to the reaction mixture under constant mechanical stirring. An orange precipitate was formed. Diluted acetic acid (0.1 M) was then added to the reaction mixture. The precipitate was filtered off, washed with water and ethanol, and recrystallized from ethanol/toluene. The yield of 1-(3-formyl-4-hydroxyphenyl azo)-4-nitrobenzene is 85%.

A solution of o-phenylenediamine (2 mmol L^−1^) in absolute ethanol (10 mL) was added to a solution of 1-(3-formyl-4-hydroxyphenyl azo)-4-nitrobenzene (4 mmoL^−1^) in absolute ethanol with stirring over 30 min at 50 °C. The mixture was heated in a water bath for 2 h at 80 °C with stirring, then cooled and left to stand at room temperature. The product was collected by filtration and washed with diethyl ether, then dried in air.

The melting point of the sample was measured using an Electro-thermal IA 9100 apparatus (Shimadzu, Japan) in open capillary tubes. The temperature at which the first drop of liquid was observed was 276 °C, as reported in [35].

### 3.4. Electrode Modification

Different concentrations of ionophore (0.3%, 0.5%, 0.7%, and 0.9% *w*/*v*) were prepared by weighing the appropriate amount (3, 5, 7, and 9 mg), dissolved in 1.0 mL ethanol and sonicated for 30 min. The GCE was modified with ionophore by dropping 1.0 µL of the NDBD solution on its surface and leaving to dry for 4 h at room temperature.

For the MWCNT/NDBD-modified electrodes, MWCNT (functionalized with carboxylate groups by concentrated nitric acid treatment using a previously described method [48]) were used with different concentrations (0.2% and 0.5% *w*/*v*) by weighing the appropriate amount (2 and 5 mg) and dissolving in 1.0 mL chitosan solution, this previously dissolved in a 1% acetic acid aqueous solution, and then sonicated for 2 h to ensure a completely homogeneous mixture. The GCE was modified by dropping 1.0 µL MWCNT and, afterwards, NDBD was drop-cast on top by the same procedure as for just NDBD, to give NDBD/MWCNT/GCE.

### 3.5. Mercury Response Measurement

The measurement of mercury ions was carried out by fixed potential amperometry. The modified electrodes were immersed into the electrochemical cell containing acetate buffer solution pH 4.0 and a fixed potential of +0.1 V vs. Ag/AgCl was applied under continuous stirring. After the steady-state current was reached, successive additions of different concentrations of Hg (II) were made, and the current response was measured as the average value registered at interval times of 0.2 s. The detection of mercury ions is based on their capture by the ionophore, schematically represented in Figure 1.

## 4. Conclusions

A new sensor for Hg^2+^ detection based on amperometric measurements, in which the ionophore NDBD was immobilized with multiwalled carbon nanotubes on a glassy carbon electrode, was developed. The NDBD/MWCNT/GCE sensor described here represents an inexpensive, fast, and simple method for the analysis of mercury ions at a low applied fixed potential. It allows the sensitive and selective amperometric determination of Hg^2+^, thus, representing a promising device for application in environmental analysis since it can be made portable and used in field measurements.

## Data Availability

The data presented in this study are available on request from the corresponding authors.

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
