# Peer review of "Novel Amperometric Mercury-Selective Sensor Based on Organic Chelator Ionophore"

_molecules, 2023, doi:10.3390/molecules28062809_

Round 1

Reviewer 1 Report

This study modified GCE with NDBD and CNT to develop an electrochemical sensor for Hg2+. The manuscript need a major revision before publication. The problems are listed as the following.

1.     The manuscript has some grammar or typing errors. For examples, “The morphology and size of the materials was investigated” (P.69),  Moreover, “ion sensitive electrodes (ISE)” (P.52) is a concept error. Generally, ISE represents ion selective electrode. Please double check the manuscript and correct all errors carefully.

2.     P. 77-78, “it is clearly observed that the tubes are thicker and several structures are visible around the tubes (indicated by the red arrows), which were attributed to iono-phore moieties. I don’t think that ionophore moieties can be seen by TEM. Is there any proof for this conclusion? I guess that these structures may be some impurities of materials.

3.     P. 88-89, “Regarding the ionophore, the XRD pattern also exhibits a wide peak around 2θ = ~ 26°, as well as two other peaks at ~ 15° and 17°,” Please explain what are these peaks seen for ionophore.

4.     In Fig. 3, the CV curves for NDBD/GCE without (black) and with (red) 2.4 mM Hg2+. Both curves show redox peaks. Please add the corresponding reaction equations. Moreover, the CV curves for Hg2+ on GCE and NDBD/CNT/GCE should be added and compared with NDBD/GCE. By the way, the papers used NDBD/GCE, NDBD/CNT and NDBD/CNT/GCE (Table 1 used NDBD/MWCNT/GCE), but only NDBD/GCE were used in Figs. 3, 4, 5, and 6. Are these results obtained without using CNT? The expression should be uniform.

5.     In Fig. 7, the amount of CNT showed much remarkable effect on the response. But I could not see the effect of NDBD. So, the result of CNT/GCE must be added and compare to demonstrate the role of NDBD for Hg2+ sensing.

6.     Please explain why the proposed sensor has an excellent selectivity for Hg2+ sensing. The selectivity experimental results showed that 3.7% Cu2+ and 6.3% Mn2+ exhibited a small interference. Why so high concentrations of 3.7% Cu2+ and 6.3% Mn2+ were used here? How about the concentration of Hg2+ in the selectivity experiments?

Author Response

This study modified GCE with NDBD and CNT to develop an electrochemical sensor for Hg2+ The manuscript need a major revision before publication. The problems are listed as the following.

  1. The manuscript has some grammar or typing errors. For examples, “The morphology and size of the materials was investigated” (P.69). Moreover, “ion sensitive electrodes (ISE)” (P.52) is a concept error. Generally, ISE represents ion selective electrode. Please double check the manuscript and correct all errors carefully.

Authors: The specific errors indicated by the reviewer were corrected. All the text was double checked for other errors by a native English speaker.

  1. 77-78, “it is clearly observed that the tubes are thicker and several structures are visible around the tubes (indicated by the red arrows), which were attributed to ionophore moieties. I don’t think that ionophore moieties can be seen by TEM. Is there any proof for this conclusion? I guess that these structures may be some impurities of materials.

Authors: By TEM it is difficult to say with certainty that it is the ionophore, and we do not have additional information; however, there are other studies which attributed similar observations of what is around the CNT to the presence of an organic compound, such as polymer [30]. We have altered the text to say that it is a possibility.

  1. 88-89, “Regarding the ionophore, the XRD pattern also exhibits a wide peak around 2θ = ~ 26°, as well as two other peaks at ~ 15° and 17°,” Please explain what are these peaks seen for ionophore.

Authors: In order to explain the peaks of the ionophore, the text has been changed as follows: “Regarding the ionophore, the XRD pattern also exhibits a wide peak around 2θ = ~ 26°, corresponding to plane reflection (002) of carbon, as well as two other peaks at ~ 15° and 17°, which may be attributed to the nitroaniline and the phenylenediamine moieties forming the ionophore’ structure [32,33].”

  1. In Fig. 3, the CV curves for NDBD/GCE without (black) and with (red) 2.4 mM Hg2+. Both curves show redox peaks. Please add the corresponding reaction equations. Moreover, the CV curves for Hg2+ on GCE and NDBD/CNT/GCE should be added and compared with NDBD/GCE. By the way, the papers used NDBD/GCE, NDBD/CNT and NDBD/CNT/GCE (Table 1 used NDBD/MWCNT/GCE), but only NDBD/GCE were used in Figs. 3, 4, 5, and 6. Are these results obtained without using CNT? The expression should be uniform.

Authors: According to the reviewer´s suggestion, the corresponding reaction equations were added. For comparison, the CVs for different electrodes were introduced in Figure 3A and the results are discussed.

The results from Table 1 and those in Figure 3, 4, 5 and 6 were obtained without MWCNT. The optimisation was carried out at the NDBD modified electrode, then, the MWCNT were incorporated in the platform in order to improve the performance. The text has been altered in order to be sure this information is clear. Thus, the heading of section 2.4 has been changed to “Analytical determination of mercury cations at modified electrodes - influence of MWCNT” and what was the first paragraph of section 2.4 has been transferred to the previous section (2.3)

  1. In Fig. 7, the amount of CNT showed much remarkable effect on the response. But I could not see the effect of NDBD. So, the result of CNT/GCE must be added and compare to demonstrate the role of NDBD for Hg2+ sensing.

Authors: Carbon nanomaterials are good alternatives to conventional adsorbents for heavy metals owing to their high surface area [41]. However, carbon nanomaterials do not show a high affinity for mercury [42], so their combination with other materials with complexing capabilities is needed, chromophores representing a good choice [27,28]. The influence of NDBD with different loadings was studied before the addition of MWCNT. Then the MWCNT were incorporated into the electrode configuration with the NDBD loading that showed better performance for mercury determination.

  1. Please explain why the proposed sensor has an excellent selectivity for Hg2+ sensing. The selectivity experimental results showed that 3.7% Cu2+ and 6.3% Mn2+ exhibited a small interference. Why so high concentrations of 3.7% Cu2+ and 6.3% Mn2+ were used here? How about the concentration of Hg2+ in the selectivity experiments?

Authors: The concentration of Hg2+ in these experiments was 3 µM and that of interferents was 6 µM, the values have been introduced in the text and in the Figure 8 caption. Generally, to check the selectivity, equal or higher concentration of interferents is used compared with that of analyte (see ref. [2, 9, 17, 18,25]).

Ionophores are chemical species that reversibly bind ions and consist of a hydrophilic centre where ions can bind surrounded by a hydrophobic part (see Scheme 1 for the one used here) and the selectivity depends on the strength of the ion-ionophore interactions. It was shown that the interaction of ion-ionophore occurs through binding to nitrogen and oxygen. Excellent selectivity for mercury was also observed in other studies based on ionophores [27,28] and may be explained considering the extraordinary stability for the N–HgII–N bonding [47].

Reviewer 2 Report

This is a well written paper, but the quality of the figures is very poor and needs improvement. The focus is new and timely. The title and abstract actually reflect what is covered in the paper. The experiments appear to have been well done as evidenced by the comprehensive and understandable results, while good scientific explanations are also written to describe all the data. Overall, the reviewer recommends that this paper be published after a minor revision.

So, I have few suggestions as follows:

 Results and discussion

Table 2.: “Data obtained from the calibration curve for Hg2+ at NDBD/CNT/GCE and NDBD/GNP/CNT/GCE……” - Where did this new modified electrode, NDBD/GNP/CNT/GCE come from, which is not mentioned anywhere before or after in the paper. If the intention was to provide data for the comparison of two differently modified electrodes, then this is not provided in this table.

            According to section 2.6., the authors demonstrated selectivity of proposed electrode towards various interfering cations or anions. However, there is no such detection result in the main figure or supplementary that reflects those hypotheses. Please clarify this issue by providing the experimental data.

Similar to section 2.6. in the section 2.7. are necessary to provide a table or a figure that will clearly show the application of the tested electrode to real samples (milk and tap water).

Materials and Methods

In chapter 3.4. the author declares the following:Different concentrations of ionophore (0.2 %, 0.5 %, 0.7 % and 0.9 % w/v) were prepared by weighing the appropriate amount (2, 5, 7 and 9 mg), dissolving in 1.0 mL ethanol and sonicating for 30 min.”, but in chapter 2.3.3. as in Table 1, the author states a concentration of 3 mg/mL, i.e. 0.3%. Please make it uniform.

Author Response

This is a well written paper, but the quality of the figures is very poor and needs improvement. The focus is new and timely. The title and abstract actually reflect what is covered in the paper. The experiments appear to have been well done as evidenced by the comprehensive and understandable results, while good scientific explanations are also written to describe all the data. Overall, the reviewer recommends that this paper be published after a minor revision.

Authors: We thank the reviewer for the suggestion. The quality of the figures has been improved.

So, I have few suggestions as follows:

 Results and discussion

  1. Table 2.: “Data obtained from the calibration curve for Hg2+at NDBD/CNT/GCE and NDBD/GNP/CNT/GCE……” - Where did this new modified electrode, NDBD/GNP/CNT/GCE come from, which is not mentioned anywhere before or after in the paper. If the intention was to provide data for the comparison of two differently modified electrodes, then this is not provided in this table.

Authors: We thank the reviewer for this observation. This was an error, “NDBD/GNP/CNT/GCE” was removed.

  1. According to section 2.6., the authors demonstrated selectivity of proposed electrode towards various interfering cations or anions. However, there is no such detection result in the main figure or supplementary that reflects those hypotheses. Please clarify this issue by providing the experimental data.

Authors: A figure that shows the results for interferents was introduced, Figure 8.

  1. Similar to section 2.6. in the section 2.7. are necessary to provide a table or a figure that will clearly show the application of the tested electrode to real samples (milk and tap water).

Authors: According to the reviewer´s suggestion, a table was introduced for the recovery measurements, Table 4.

  1. Materials and Methods

In chapter 3.4. the author declares the following: „Different concentrations of ionophore (0.2 %, 0.5 %, 0.7 % and 0.9 % w/v) were prepared by weighing the appropriate amount (2, 5, 7 and 9 mg), dissolving in 1.0 mL ethanol and sonicating for 30 min.”, but in chapter 2.3.3. as in Table 1, the author states a concentration of 3 mg/mL, i.e. 0.3%. Please make it uniform.

Authors: We thank the reviewer for this observation. The error was corrected in section 3.4.

Reviewer 3 Report

In this article, the authors developed a new amperometric sensor based on the organic chelating ionophore N,Ndi(2-hydroxy-5-[(4-nitrophenyl)diazenyl]benzalde-11hide)benzene-1,2-diamine (NDBD) and nanotubes of carbon (CNT) immobilized on a glassy carbon for the determination of toxic mercury ions in milk and water samples. The article presents a novelty in terms of the construction method of the new sensor, but the detection of heavy metals in different samples has been intensively studied in recent years as well. I think that the study could have been much more interesting if the detection of more heavy metals had been pursued or if there had been more applications. However, the innovative method of preparing the nanocompound could add value to the scientific world.

Therefore, I consider that the article could be published in the journal Molecules, only after fixing some rather important aspects.

1. In section 3.5 you could add more details regarding the electrochemical measurements, for example the electrochemical detection method, the working parameters.

2. In figure 2B - you can highlight on the figure the main bands that make the difference between the nanocompounds, so that it is clearer for the readers.

3. In section 2, an electrochemical characterization method before adding HG2+ would be useful, such as EIS to determine the resistance to electron transfer to the modified sensors.

4. The resolution of the figures is weak. It would be useful if the size of the figures were increased.

5. Section 2.2. A comparison between all modified sensors in the presence/absence of Hg2+ is missing. I suggest adding a figure that includes the CV for each sensor.

6. In section 2.3.1.- Add a more eloquent and complex explanation for the electrochemical process that takes place on the surface of the electrode, regarding the decrease in current intensity starting with +0.3 (possibly with a corresponding reference).

7. Please explain why you did not choose +0.2 V instead of +0.1 V as a fixed potential, even if the increase in response was not as intense.

8. Section 2.3.3 – add 1-2 citations to support the choice of pH. The detection of Hg2+ and heavy metals in general is intensively studied, and the pH in the literature, in many works, was 4.5 and even 5.

9. In section 2.4, please add a figure that contains the equation of the line and the value of R2 or add this information to Table 1, possibly also the LOQ.

10. It is not clear whether it is carbon nanotubes or multilayer carbon nanotubes. I find different notations regarding the carbon nanomaterial and the name of the sensor. Ex: table 3 appears NDBD/MWCNT/GCE and in the legend of figure 7 appears NDBD/CNT/GCE.

11. In section 2.6. add a table or figure to confirm these results.

12. Also, in section 2.7, no table or figure reveals results. Please add for better clarity and ease of reading.

13. I suggest adding some newer references. Only 10 references out of 58 are from the last 5 years.

Author Response

In this article, the authors developed a new amperometric sensor based on the organic chelating ionophore N,Ndi(2-hydroxy-5-[(4-nitrophenyl)diazenyl]benzalde-11hide)benzene-1,2-diamine (NDBD) and nanotubes of carbon (CNT) immobilized on a glassy carbon for the determination of toxic mercury ions in milk and water samples. The article presents a novelty in terms of the construction method of the new sensor, but the detection of heavy metals in different samples has been intensively studied in recent years as well. I think that the study could have been much more interesting if the detection of more heavy metals had been pursued or if there had been more applications. However, the innovative method of preparing the nanocompound could add value to the scientific world.

Therefore, I consider that the article could be published in the journal Molecules, only after fixing some rather important aspects.

  1. In section 3.5 you could add more details regarding the electrochemical measurements, for example the electrochemical detection method, the working parameters.

Authors: Following the reviewer´s suggestion, more details regarding the electrochemical measurements were added.

  1. In figure 2B - you can highlight on the figure the main bands that make the difference between the nanocompounds, so that it is clearer for the readers.

Authors: According to the reviewer´s suggestion, the main bands that differentiate the nanocompounds are now highlighted in Figure 2B.

  1. In section 2, an electrochemical characterization method before adding Hg2+ would be useful, such as EIS to determine the resistance to electron transfer to the modified sensors.

Authors: EIS measurements had already been made with the NDBD/GCE and NDBD/MWCNT/GCE in electrolyte solution. Unfortunately, these spectra did not differ much in their profile or impedance magnitudes. It does suggest that the surface active area is similar in all cases.
Note that spectra are often recorded in the presence of potassium hexacyanoferrate, from which a value of the charge transfer resistance can be deduced that simply depends on the available active surface area of the modified electrode available for hexacyanoferrate oxidation/reduction. However, it is a foreign species (of negative charge) that could modify the electrode response to analyte and it is the hexacyanoferrate response that is characterised rather than the modified electrode directly.

  1. The resolution of the figures is weak. It would be useful if the size of the figures were increased.

Authors: We thank the reviewer for this comment. The resolution of the figures was increased, as much as possible given the restrictions of the template.

  1. Section 2.2. A comparison between all modified sensors in the presence/absence of Hg2+ is missing. I suggest adding a figure that includes the CV for each sensor.

Authors: A new figure has been introduced, Figure 3A, which includes CVs at different electrodes in the presence of Hg2+.

  1. In section 2.3.1.- Add a more eloquent and complex explanation for the electrochemical process that takes place on the surface of the electrode, regarding the decrease in current intensity starting with +0.3 (possibly with a corresponding reference).

Authors: The following has been written. “The decrease in current response by amperometry occurs at a slightly lower potential than the peak observed by cyclic voltammetry. This is not unexpected, since CV is dynamic so that the fixed potential amperometry peak will correspond to a CV obtained at low scan rate. As observed in Figure 3B, when the scan rate is increased, the CV peak potentials shift more apart. The response was the highest at +0.2 V, an increase in current of 30% occurring from 0.0 V to +0.1 V; from +0.1 V to +0.2 V the current increased by a further 10%, which can be correlated with the ionophore attracting Hg2+ and making a stable complex. A possible explanation of the decrease at higher potentials from +0.3 V onwards may be that the ionophore complex is less stable.”

  1. Please explain why you did not choose +0.2 V instead of +0.1 V as a fixed potential, even if the increase in response was not as intense.

Authors: An explanation for this choice was added “The reason for choosing a potential that is closer to 0 V is to reduce the effect of possible interferences from other ions, which undergo reaction around +0.2 - 0.3 V, e.g. Cu(II) [39].”

  1. Section 2.3.3 – add 1-2 citations to support the choice of pH. The detection of Hg2+ and heavy metals in general is intensively studied, and the pH in the literature, in many works, was 4.5 and even 5.

Authors: References were added to support the choice of pH. “The decrease above pH 4.0 may be attributed to the formation of hydroxy complexes of Hg (II), as also observed in other studies, although the pH at which this occurs varies according to the modifier used. For example, in [1] this decrease is observed at pH higher than 4.0, in [25] it happens above pH 3.0, while in [18] this was only seen at pH values higher than 7.0. On the other hand, lower pH values lead to competition between mercury ions and protons [1].”

  1. In section 2.4, please add a figure that contains the equation of the line and the value of R2 or add this information to Table 1, possibly also the LOQ.

Authors: According to the reviewer´s suggestion, we have added the required information in Table 1 and Table 2.

  1. It is not clear whether it is carbon nanotubes or multilayer carbon nanotubes. I find different notations regarding the carbon nanomaterial and the name of the sensor. Ex: table 3 appears NDBD/MWCNT/GCE and in the legend of figure 7 appears NDBD/CNT/GCE.

Authors: The nanotubes used were multiwalled carbon nanotubes (MWCNT). The abbreviation MWCNT is now used throughout the text, except in the more general text in the introduction.

  1. In section 2.6. add a table or figure to confirm these results.

Authors: Figure 8 was introduced with the corresponding results.

  1. Also, in section 2.7, no table or figure reveals results. Please add for better clarity and ease of reading.

Authors: As requested, a table, Table 4, has been introduced with the recovery measurements.

  1. I suggest adding some newer references. Only 10 references out of 58 are from the last 5 years.

Authors: According to reviewer´s suggestion, over 20 older references were removed and/or were replaced by more recent references.

Round 2

Reviewer 1 Report

 This revised version is acceptable.

Reviewer 3 Report

The article has been improved considerably. I believe that it can be published in Molecules journal in its present form.